# Habitual Physical Activity and Dietary Profiles in Older Japanese Males with Normal-Weight Obesity

**DOI:** 10.3390/ijerph20146408

**Published:** 2023-07-20

**Authors:** Yusei Tataka, Ayano Hiratsu, Kyoko Fujihira, Chihiro Nagayama, Kayoko Kamemoto, Takashi Fushimi, Hideto Takase, Masashi Miyashita

**Affiliations:** 1Graduate School of Sport Sciences, Waseda University, 2-579-15 Mikajima, Tokorozawa, Saitama 359-1192, Japan; 2Waseda Institute for Sport Sciences, Waseda University, 2-579-15 Mikajima, Tokorozawa, Saitama 359-1192, Japan; 3Biological Science Research, Kao Corporation, 2-1-3 Bunka, Sumida-ku, Tokyo 131-8501, Japan; 4Faculty of Sport Sciences, Waseda University, 2-579-15 Mikajima, Tokorozawa, Saitama 359-1192, Japan; 5School of Sport, Exercise and Health Sciences, Loughborough University, Epinal Way, Loughborough, Leicestershire LE11 3TU, UK; 6Department of Sports Science and Physical Education, The Chinese University of Hong Kong, Shatin 999077, Hong Kong

**Keywords:** visceral fat area, normal-weight obesity, older males, physical activity, diet

## Abstract

Normal-weight obesity is defined as having high body fat but a normal body mass index (BMI). This study examined whether there are differences in habitual physical activity and diet between individuals with normal-weight obesity and obese or non-obesity. This study included 143 males aged 65–75 years, and they were classified into the following three groups according to BMI and visceral fat area (VFA): obese group (n = 27 (BMI: ≥25 kg/m^2^ and VFA: ≥100 cm^2^)), normal-weight obese group (n = 35 (BMI: <25 kg/m^2^ and VFA: ≥100 cm^2^)) and non-obese group (n = 81 (BMI: <25 kg/m^2^ and VFA < 100 cm^2^)). Lowered high-density lipoprotein cholesterol and elevated triglyceride and alanine transaminase were observed in the normal-weight obese group than in the non-obese group (all for *p* ≤ 0.04, effect size ≥ 0.50). No differences were found in physical activity and dietary habits between non-obese and normal-weight obese groups (all for *p* > 0.05). Although impaired lipid and liver function parameters were observed in older males with normal-weight obesity compared with older males with non-obesity, physical activity and dietary profiles in themselves were not shown these differences in the present study.

## 1. Introduction

Obesity represents a leading public health issue worldwide today [1]. Adipose tissue in the body is distributed into two main categories: subcutaneous and visceral adipose tissue. Visceral fat accumulation is independently associated with elevated risks of type 2 diabetes and dyslipidaemia [2,3] or more strongly associated with metabolic diseases than subcutaneous fat [4]. Although body mass index (BMI) is commonly used to determine obesity, it has been demonstrated that BMI and proxy indicators of excess fat accumulation, such as waist circumference, are largely ineffective in identifying visceral fat obesity [5,6]. Given the effect of visceral fat accumulation on health and the measurement limitations of BMI, some attempts have been made to better identify obesity by using measures of both BMI and body fat percentage or visceral fat area. In Japan, a “BMI < 25 kg/m^2^ and visceral fat area (cm^2^) ≥ 100” is used as a reference value, and a population meeting this criterion is considered normal-weight obesity [7]. Normal-weight obesity is defined as having high body fat but a normal BMI [8,9].

Previous studies have demonstrated that even after adjusting for total body fat mass, females had a lower ratio of visceral fat tissue in total body fat mass than males [10], and at the equivalent levels of waist circumference, males had larger areas of visceral fat tissue than females [11]. Premenopausal females are more likely to store lipids as subcutaneous fat in the lower body due to hyperplasia of adipocytes, while males and postmenopausal females are more prone to deposit visceral adipose tissue in the abdominal region through adipocyte hypertrophy [12]. It is well recognised that aging promotes visceral adipose tissue accumulation as it is accompanied by decreased testosterone in males [13]. Furthermore, ethnicity influences the risk of visceral adipose tissue accumulation [12]. Asians are more predisposed to visceral fat than Europeans and Caucasian Americans, despite having lower total adiposity [14]. Japanese people are also more likely to accumulate visceral fat tissue than African-Americans and Caucasians [15] since the size of the preadipocyte pool of subcutaneous adipose tissue is small [16].

It has been suggested that overconsumption of foods and/or lower levels of physical activity often links to excessive accumulation of visceral fat [12]. These associations were proved by some lifestyle modifications, including nutrition and physical activity interventions [17]. A reduction and a loss of visceral fat tissue could be observed even in the absence of body mass loss, as the advantage of vigorous habitual exercise over energy restriction is that it even increases or could preserve lean muscle mass [12]. However, limited studies have examined which diet and physical activity habits influence visceral and fat accumulation [18,19], and the characteristics of these lifestyles in individuals with normal-weight obesity remain unclear. Since individuals with normal-weight obesity have several risks of lifestyle diseases, including insulin resistance, subclinical cardiovascular disease, and metabolic abnormalities [20], it is important to examine the characteristics of lifestyle factors associated with visceral fat accumulation in preventing metabolic morbidity. Differences in physical activity and dietary characteristics between individuals with overweight and obesity and individuals with normal weight have been elucidated by previous cross-sectional and cohort studies: physical activity: Refs. [21,22,23,24]; diet: Refs. [24,25,26,27]. Therefore, the aim of the present study was to clarify these characteristics of older Japanese males with normal-weight obesity by adding normal-weight obesity to individuals with obesity and normal weight, for whom differences in physical activity and diet have already been identified.

## 2. Materials and Methods

### 2.1. Participants

A participant flow diagram is presented in Figure 1. The dataset used in the present study was from our previous study [28]. Th present study was conducted with the approval of the Institutional Ethics Committee on Human Research (approval number: 2016-290) and was conducted in accordance with the Declaration of Helsinki. All participants provided written informed consent to participate in this study. The present study was registered in advance with the University Hospital Medical Information Network Center (UMIN), a system for registering clinical trials (ID: UMIN000026007). All participants were fully informed of the nature and purpose of the study and provided written consent to participate. Inclusion criteria included Japanese older adults 65–75 years of age who are able to take photos with a disposal camera regarding all consumed meals for 3 days during the study, can wear accelerometers for one week during the study and can follow the instructions by the experimenters and medical staff. Initially, 490 participants were enrolled. Among 469 participants who completed the study [28], 143 males were selected for the analysis after excluding 23 males who had missing data in this study. Then, the participants were allocated into the three groups according to their BMIs and visceral fat areas: (1) obese group (n = 27 (BMI: ≥25 kg/m^2^ and visceral fat area: ≥100 cm^2^)), (2) normal-weight obese group (n = 35 (BMI: <25 kg/m^2^ and visceral fat area: ≥100 cm^2^)) and (3) non-obese group (n = 81 (BMI: <25 kg/m^2^ and visceral fat area <100 cm^2^)).

### 2.2. Anthropometric Measurements

The value of body mass in kilogrammes (Inner Scan 50, Tanita Corporation, Tokyo, Japan) divided by the square of height in metres (YS-OA, AS ONE Corporation, Osaka, Japan) was used to calculate body mass index. Arterial blood pressure was measured from the left arm (605P, Yagami Co., Ltd., Yokohama, Japan). Blood pressure measurements were taken twice in total, and the mean of these values was calculated. Visceral fat was measured using abdominal bioelectrical impedance analysis, which has four electrodes on a belt (EW-FA90, Panasonic, Osaka, Japan) [29,30]. The device was placed at a participant’s umbilical level while the participant was in a standing position. Waist circumference was measured while the participant was in a standing position at the level of the umbilicus using a constant tension tape.

### 2.3. Physical Activity Measurements

Participants wore a uniaxial accelerometer (Lifecoder-EX, Suzuken Co., Ltd., Aichi, Japan) to evaluate their physical activity levels, including a total step count (steps/day) during the study period. The accelerometer defines 11 levels of physical activity intensity (0, 0.5, and 1–9), with 0 being the lowest activity intensity and 9 being the highest activity intensity. Physical activity intensity was classified as light intensity (level of activity intensity 1–3), moderate intensity (level of activity intensity 4–6) and vigorous intensity (level of intensity 7–9). Level 4 corresponds to an intensity of approximately 3 metabolic equivalents [31]. Furthermore, total step was calculated using software (Lifelyzer 05 Coach, Suzuken Co., Ltd., Aichi, Japan). At least four days of recording (over three weekdays and one weekend day), with a minimum of 10 h per day of wear time, was required for inclusion in the analysis [32].

### 2.4. Physical Fitness Measurements

The present study assessed the participant’s physical function by conducting the following physical function tests; grip strength, 20-s Open–Close Stepping Test, Functional Reach Test, knee strength (per body mass), one-leg standing with eyes opened and Timed Up and Go Test. For detailed information regarding each measurement, refer to our previous study [28]. Regarding the 20-s Open–Close Stepping Test, given the characteristics of the target population and the objectives of the study, this test was used in the present study to assess agility function as it can be performed safely and conveniently in a sitting position.

### 2.5. Dietary Record

Participants took a photo of all meals, including beverages and snacks, for three consecutive days (i.e., Thursday, Friday and Saturday) starting on a non-working day. When taking photographs of each meal, the participants were asked to include standard-size tableware in the same photograph so that the size of foodstuffs could be estimated [33]. The items consumed, meal duration, approximate amount and location were also recorded. All meals and beverages consumed during the study period were recorded and photographed using a dietary chart and a camera (Utsurundesu, FUJIFILM Corporation, Tokyo, Japan) by participants. The photographs of meals were analysed for nutrients and food groups according to the Standard Tables of Food Composition in Japan [34] by a registered dietitian. The analysis was conducted using the software (Healthy Maker Pro 501, Mushroom Soft, Co., Ltd., Okayama, Japan), which contains a database of foods and meals commonly consumed by Japanese adults. Where recorded items by the participants were not listed in the tables [34], they were replaced with similar items.

### 2.6. Analytical Methods

Venous blood samples were obtained from the arm vein in a fasted state. Whole blood was obtained in sodium fluoride–ethylenediaminetetraacetic acid tubes, and these were used to measure plasma glucose and hemoglobin A1c (HbA1c). Whole blood was obtained in dipotassium salt–ethylenediaminetetraacetic acid tubes, and these were used to measure plasma insulin. Both tubes were immediately centrifuged at 3000 rotations per minute (rpm) for 10 min at 4 °C. The plasma was then dispensed into plain microtubes and stored at −80 °C until analysis. Whole blood was obtained in dipeptidyl peptidase 4 inhibitor and protease inhibitor cocktail (BD, Tokyo, Japan) tubes, and these tubes were immediately centrifuged and treated as described above. These treated samples were used to measure plasma glucose-dependent insulinotropic polypeptide (GIP). Whole blood was obtained in clotting activator tubes for isolation of serum, and these tubes were allowed to clot for 30 min at room temperature and then centrifuged as described above. These treated samples were used to measure for serum high-density lipoprotein cholesterol (HDL-C), low-density lipoprotein cholesterol (LDL-C), triglycerides (TG), non-esterified fatty acids (NEFA), total protein (TP), albumin, uric acid, aspartate aminotransferase (AST), alanine aminotransferase (ALT) and γ-glutamyl transpeptidase (γ-GTP). Enzymatic colorimetric assays were used to measure the plasma concentration of glucose (GLU-HK (M), Shino-test Corporation, Kanagawa, Japan), serum LDL-C (MetaboLead LDL-C, Hitachi Chemical Diagnostics Systems Co., Ltd., Tokyo, Japan), serum HDL-C (MetaboLead HDL-C, Hitachi Chemical Diagnostics Systems Co., Ltd., Tokyo, Japan), serum TG (Pure-Auto S TAG-N, Sekisui Medical Company Limited, Tokyo, Japan) and serum NEFA (NEFA-HR, Wako Pure Chemical Industries Limited, Osaka, Japan). Plasma concentrations of insulin (Mercodia Insulin ELISA, Mercodia AB, Uppsala, Sweden), HbA1c (MetaboLeadHbA1c, Minaris Medical Co., Ltd., Tokyo, Japan) and total GIP (EMD Millipore ELISA, EMD Millipore Corporation, Missouri, USA) were measured by enzyme-linked immunosorbent assay. Plasma concentration of albumin (L-type Wako ALB-BCP, Wako Pure Chemical Industries Limited, Osaka, Japan) was measured by the modified bromocresol purple assay. Serum uric acid (Detaminar C-UA, Minaris Medical Co., Ltd., Tokyo, Japan) was measured by uricase and peroxidase method assay. Serum TP concentration was determined by the Biuret method (TP-HR, FUJIFILM Wako Pure Chemical Corporation, Osaka, Japan). Serum AST (L-type Wako AST J2, Wako Pure Chemical Industries Limited, Osaka, Japan), serum ALT (L-type Wako ALT J2, Wako Pure Chemical Industries Limited, Osaka, Japan) and serum γ-GTP (L-type Wako γ-GT J, Wako Pure Chemical Industries Limited, Osaka, Japan) were measured by Japan Society of Clinical Chemistry transferable method. Homeostasis model assessment of insulin resistance (HOMA-IR) index was calculated as follows: fasting insulin (μIU/mL) × fasting glucose (mmol/L)/22.5 [35]. Homeostasis model assessment of β-cell function (HOMA-β) index was calculated as follows: 20 × fasting insulin (μIU/mL)/(fasting glucose (mmol/L) − 3.5) [36]. A higher HOMA-IR represents reduced insulin sensitivity, and a lower HOMA-β represents decreased fasting insulin secretion [36,37,38].

### 2.7. Statistical Analyses

Data were analysed with Predictive Analytics Software version 28.0 for Windows (IBM Corporation, New York, NY, USA). Normality of the data was checked using Shapiro–Wilk tests. Data that were not normally distributed were naturally log-transformed. Physical characteristics, blood parameters, physical activity data and dietary data were compared among the non-obese, normal-weight obese and obese groups using generalised estimating equations with group included as a fixed factor. Analyses of blood parameters were adjusted for total physical activity, and analyses of physical activity and diet were adjusted for age. Where a significant group effect was identified, the data were subsequently analysed using post-hoc analysis and were adjusted for multiple comparisons using the Bonferroni method. We did not adjust for *p*-value for multiple comparisons because the number of pair-wise comparisons was relatively small in the present study. To complement the findings, absolute standardised effect sizes (*d*) (Cohen’s d) are presented. For all outcome measures, an effect size of 0.2 was considered a small difference, 0.5 moderate and 0.8 large [39]. Statistical significance was accepted at the 5% level. Results are reported as the mean ± standard deviation.

## 3. Results

### 3.1. Physical Characteristics

Differences were observed in body mass, BMI and visceral fat area among groups (Table 1; all for *p* < 0.01). Post-hoc analyses of the main effect of the group showed that body mass was higher in the obese group than in the normal-weight obese (*d* = 1.61) and non-obese groups (*d* = 2.38) (all for *p* < 0.01) and was higher in the normal-weight obese group than the non-obese group (*p* < 0.01, *d* = 0.74). Post-hoc analyses of the main effect of the group showed that BMI was higher in the obese group than in the normal-weight obese (*d* = 2.47) and non-obese groups (*d* = 3.18) (all for *p* < 0.01) and was higher in the normal-weight obese group than the non-obese group (*p* < 0.01, *d* = 0.80). Post-hoc analyses of the main effect of the group showed that visceral fat area was higher in the obese group than in the normal-weight obese (*d* = 0.81) and non-obese groups (*d* = 3.22) (all for *p* < 0.01) and was higher in the normal-weight obese group than the non-obese group (*p* < 0.01, *d* = 2.85). No differences were found in age (*p* = 0.747), height (*p* = 0.683), systolic blood pressure (*p* = 0.572) or diastolic blood pressure (*p* = 0.732) among groups. Differences were observed in the one-leg standing with vision among groups (*p* < 0.01). Post-hoc analyses of the main effect of the group showed that the time of one-leg standing with vision was longer in the non-obese group than in the normal-weight obese (*d* = 0.68) and obese groups (*d* = 0.53) (all for *p* < 0.05). No differences were found among groups in grip strength (*p* = 0.296), knee strength (per body mass) (*p* = 0.606), TUG (*p* = 0.788), OCS-10 (*p* = 0.569) or FRT (*p* = 0.997) among groups.

### 3.2. Blood Parameters

Differences were observed in circulating concentrations of glucose, insulin, HbA1c and HOMA-IR among groups (Table 2; all for *p* < 0.05). Post-hoc analyses of the main effect of the group showed that circulating concentrations of glucose (*d* = 0.67), insulin (*d* = 0.75) and HbA1c (*d* = 0.60) were higher in the obese group than in the non-obese group (all for *p* < 0.05). Post-hoc analyses of the main effect of the group showed that plasma HOMA-IR was higher in the obese group than in the normal-weight obesity (*d* = 0.65) and the non-obese group (*d* = 0.87) (all for *p* < 0.05). No difference was found in HOMA-β and plasma GIP concentration among groups (*p* = 0.073, *p* = 0.354). Differences were observed in serum HDL-C and TG concentrations among groups (all for *p* < 0.05). Post-hoc analyses of the main effect of the group showed that serum HDL-C was higher in the non-obese group than in the normal-weight obese groups (*p* < 0.05, *d* = 0.75). Post-hoc analyses of the main effect of the group showed that serum TG was higher in the normal-weight obese group than in the non-obese group (*p* < 0.05, *d* = 0.50). No differences were found in serum LDL-C (*p* = 0.392), NEFA (*p* = 0.685), TP (*p* = 0.086), albumin (*p* = 0.384), uric acid (*p* = 0.164), γ-GTP (*p* = 0.513) or AST (*p* = 0.832), concentrations among groups. Differences were observed in serum ALT concentrations among groups (*p* < 0.05). Post-hoc analyses of the main effect of the group showed that serum ALT was lower in the non-obese group than in the normal-weight obese (*d* = 0.57) and obese groups (*d* = 0.88) (all for *p* < 0.05).

### 3.3. Physical Activity

Differences were observed in step count, total physical activity and vigorous-intensity physical activity among groups (Table 3; all for *p* < 0.05). Post-hoc analyses of the main effect of the group showed that step count and total physical activity were higher in the non-obese group than in the obese group (*p* < 0.05; *d* = 0.63, 0.61). Post-hoc analyses of the main effect of the group showed that vigorous-intensity physical activity was higher in the non-obese group than in the normal-weight obese and obese groups (all for *p* > 0.05, *d* ≥ 0.28). No differences were found in light-intensity (*p* = 0.114) or moderate-intensity (*p* = 0.068) physical activity among groups.

### 3.4. Diet

Differences were observed in the consumption of grains and vegetable food groups among groups (Table 4; all for *p* < 0.05). Post-hoc analyses of the main effect of the group showed that the consumption of grain and vegetable food groups was lower in the obese group than in the non-obese (all for *d* ≥ 0.58) and normal weight groups (all for *d* ≥ 0.67) (all for *p* < 0.05). No differences were found in energy intake per day (i.e., breakfast (*p* = 0.466), lunch (*p* = 0.413), dinner (*p* = 0.190), snack (*p* = 0.172), total energy intake (*p* = 0.523)), macronutrient ratio (i.e., protein (*p* = 0.276), fat (*p* = 0.373) and carbohydrate (*p* = 0.441)) or every food groups except grains and vegetables food groups among groups (all for *p* > 0.05).

## 4. Discussion

The present study found that individuals with normal-weight obesity exhibited elevated TG and ALT and low HDL-C concentrations compared with non-obese individuals, but these did not appear to be influenced by physical activity or diet.

Differences in lipid and liver function parameters were found between the normal-weight obese and non-obese groups, with higher concentrations of TG and ALT and lower concentrations of HDL-C in the normal-weight obese group compared to the non-obese group. It has been suggested that hypertriglyceridaemia and low HDL-C are two main detectable blood abnormalities that are associated with visceral obesity in a typical clinical setting [12]. Also, the previous study has shown that the visceral fat area was strongly associated with circulating ALT activities, even though this association was independent of the subcutaneous fat area [40]. Furthermore, high plasma TG, low plasma HDL-C and ALT concentrations are related to insulin resistance [41,42]. Indeed, it has been reported that there was a correlation between HOMA-IR and visceral fat mass [43], and this was also the case in the present study (r = 0.253, *p* < 0.001). Moreover, a previous study that assessed visceral fat area using the bioelectrical impedance analysis in 2336 middle-aged (i.e., 48.0 ± 10.5 years old, mean ± standard deviation) Japanese males reported that the normal-weight obesity group (n = 401/2336 (17%), visceral fat area ≥100 cm^2^ and BMI <25 kg/m^2^) exhibited more metabolic risk factors such as dyslipidaemia, dysglycaemia/impaired glucose tolerance or elevated blood pressure than the overall obese group without visceral fat accumulation (n = 98/2336 (4%), visceral fat area <100 cm^2^ and BMI ≥25 kg/m^2^) [44]. In addition, this study reported that a reduction in visceral fat over one year was associated with a significant reduction in the number of metabolic risks when participants who received new treatment within the same period were excluded from the analysis [44]. Thus, it is important to assess visceral fat area in addition to BMI, at least from the perspective of preventing metabolic syndrome, as visceral fat accumulation is associated with the clusters of metabolic risk factors, regardless of BMI.

A previous cross-sectional study has shown that reduced regular physical activity was associated with visceral fat accumulation [45]. Furthermore, the previous cross-sectional study with the use of isotemporal substitution analyses to estimate the effect of replacing sitting time with physical activity has reported that replacing 30 min per day of sitting time with 30 min per day of moderate-to-vigorous physical activity was associated with the reduction of visceral fat [46]. However, there was no difference in total physical activity, low-intensity physical activity or moderate-intensity physical activity between the normal-weight obese and non-obese groups in the present study. It is interesting to note that the normal-weight obese group achieved the recommended amount of moderate physical activity (i.e., ≥150 min/week) set by the World Health Organisation in the present study (normal-weight obese group; 162 min/week, non-obese group; 217 min/week; obese group; 147 min/week). The previous study has reported that South Asian males may need to engage in more amounts of moderate physical activity to exhibit a cardio-metabolic risk profile similar to Europeans, suggesting ethnicity-specific physical activity guidelines [47]. To corroborate this concept, a recent review has reported that the effectiveness of physical activity on carbohydrate and lipid metabolism parameters (i.e., glucose, insulin and TG) is influenced by ethnicity, and this difference is partly mediated by visceral fat accumulation, which impairs insulin sensitivity and pancreatic β cell function [48]. Therefore, it is necessary to examine which physical activity practices are useful for the prevention of visceral fat accumulation in order to develop ethnicity-specific prevention and management guidelines for visceral fat accumulation through physical activity.

A previous meta-analysis that examined the effect of caloric restriction on visceral adipose tissue loss in individuals with overweight and obesity has reported that caloric restriction reduces visceral adipose tissue [17]. A previous longitudinal study of middle-aged Japanese males and females showed that changes in the visceral fat area are negatively associated with the consumption of pantothenic acid soluble dietary fiber, potassium, manganese, magnesium, folic acid and vitamin K and positively with the consumption of monounsaturated fat [49]. Meanwhile, in the baseline measurements of the same previous study as above, only retinol intake was negatively correlated with visceral fat area in males when divided into two groups, above or less than 100 cm^2^ [49]. Moreover, a systematic review of the evidence from cross-sectional and controlled intervention studies on the association between visceral adipose tissue and qualitative aspects of diet reported that visceral adipose tissue was inversely associated with medium-chain TG, dietary fibre, calcium and various phytochemicals [19]. In contrast to these previous findings, no differences in quantitative and qualitative aspects of the diet were found between normal-weight obese and non-obese groups in the present study. The reasons for the inconsistent results between the studies on the relationship between visceral fat area and quantitative and qualitative aspects of diet, including the results of the present study, may be due to methodological differences between the studies, such as the target population (i.e., age, sex and ethnicity), and the methods or techniques used to assess diet (i.e., 24-h recall, Brief Diet History Questionnaire and Food Frequency Questionnaire) and visceral fat area (i.e., computed tomography, magnetic resonance imaging).

The primary aim of our study was to simply compare the physical activity and dietary data among the three groups. Differences in physical activity and dietary characteristics between individuals with overweight or obesity and individuals with normal weight have been elucidated by previous cross-sectional and cohort studies [21,22,23,24,25,26,27]. On the other hand, the characteristics of physical activity and diet in individuals with normal-weight obesity remain unclear. Therefore, our study aimed to clarify these characteristics of individuals with normal-weight obesity by adding individuals with normal-weight obesity to individuals with obesity and normal-weight, for whom differences in physical activity and diet have already been identified. Previous studies have reported significant relationships between total physical activity and grip strength [50] and between diet (total intake of various items, including nuts, pulses, fruits, vegetables, mollusks, crustaceans and fish) and insulin [51]. The present study did not examine and report correlations between the parameters reported in each table for the following reasons: (1). The main aim of the present study was to simply compare physical activity and dietary data among the three groups as addressed; and (2). The number of participants recruited for the present study was not sufficient to conduct correlation analyses. Nonetheless, further large studies need to consider these correction analyses for our understanding of the characteristics of physical activity and diet in individuals with normal-weight obesity.

One strength of our study was the quantification of physical activity levels using objective data obtained from accelerometer measurements, as self-report measures are limited in their ability to accurately quantify physical activity levels due to poor recall of activities and frequent overestimation of physical activity levels [52]. A major limitation of our study was a small sample size which made us unable to stratify the obesity group without visceral fat accumulation (BMI: ≥25 kg/m^2^ and visceral fat area: <100 cm^2^). Additionally, the method in the present study to predict visceral fat was measured by bioelectrical impedance analysis. Although the previous study reported that the visceral fat area presumed by abdominal bioelectrical impedance analysis correlated with the visceral fat area determined by computed tomography [29], conventional methods, including computed tomography or magnetic resonance imaging, are needed to employ for comparison among studies. In addition, since the participants of the present study were older males, the findings of the present study may be readily generalised to older females or other age groups. Furthermore, our data were collected from relatively active participants who lived in urban areas. Thus, data from other geographical areas are needed to determine habitual physical activity and dietary profiles in individuals with normal-weight obesity.

## 5. Conclusions

In conclusion, older Japanese males with increased visceral fat area but normal BMI have exhibited impaired lipid and liver function parameters compared with older Japanese males with non-obesity in the present study. However, lifestyle factors, including dietary patterns and physical activity levels themselves, were not shown these differences among groups with different amounts of visceral fat area and BMI in the present study.

## Figures and Tables

**Figure 1 ijerph-20-06408-f001:**
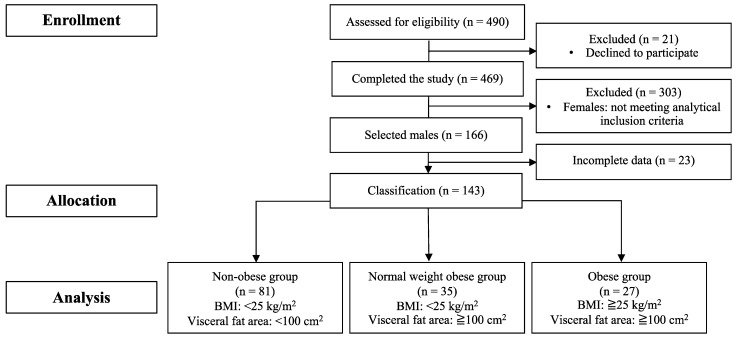
The flow diagram of participants through the study.

**Table 1 ijerph-20-06408-t001:** The characteristics of participants in the obese, normal-weight obese and non-obese groups.

	Obese Group (n = 27)	Normal-Weight Obese Group (n = 35)	Non-Obese Group (n = 81)	Main Effect of Group (*p* Value)
Age (year)	70.0 (3.3)	70.0 (2.9)	70.4 (2.9)	NS
Height (m)	1.66 (0.07)	1.68 (0.05)	1.67 (0.05)	NS
Body mass (kg)	75.6 (7.9) *^,†^	65.0 (5.2) *	61.1 (5.3)	<0.05
Body mass index (kg/m^2^)	27.3 (2.0) *^,†^	23.1 (1.3) *	21.9 (1.6)	<0.05
Visceral fat area (cm^2^)	149.7 (36.8) *^,†^	125.9 (20.8) *	63.8 (22.0)	<0.05
Systolic blood pressure (mmHg)	146 (21)	143 (28)	141 (22)	NS
Diastolic blood pressure (mmHg)	83 (12)	84 (13)	82 (12)	NS
Grip strength (kg)	36.2 (7.6)	33.2 (8.6)	34.1 (7.0)	NS
Functional Reach Test (cm)	35.4 (6.7)	35.4 (6.2)	35.5 (6.4)	NS
OCS-20 (count)	29 (6)	29 (5)	30 (5)	NS
Knee strength (kgf/kg)	0.41 (0.12)	0.40 (0.15)	0.40 (0.10)	NS
One-leg standing with vision (s)	57 (40) *	50 (44) *	81 (46)	<0.05
Timed Up and Go Test (s)	5.59 (1.10)	5.61 (0.89)	5.64 (0.83)	NS

Values are means (standard deviation). Means were compared using the generalised estimating equations. Post-hoc analysis was adjusted for multiple comparisons using the Bonferroni method. Post-hoc analysis of the main effect of group: * *p* < 0.05 different from the non-obese group. ^†^
*p* < 0.05 different from the normal-weight obese group. OCS-20; 20-s Open–Close Stepping Test, NS; Not significant.

**Table 2 ijerph-20-06408-t002:** Blood parameters in the obese, normal-weight obese and non-obese groups.

	Obese Group (n = 27)	Normal-Weight Obese Group (n = 35)	Non-Obese Group (n = 81)	Main Effect of Group (*p* Value)
Glucose (mmol/L)	6.2 (1.1) *	5.8 (1.0)	5.6 (0.8)	<0.05
HbA1c (%)	5.9 (0.7) *	5.7 (0.5)	5.6 (0.4)	<0.05
Insulin (pmol/L)	58.3 (54.6) *	35.6 (16.8)	31.9 (25.5)	<0.05
HOMA-IR	2.4 (2.5) *^,†^	1.3 (0.6)	1.1 (1.0)	<0.05
HOMA-β	61.1 (44.4)	46.8 (21.3)	43.8 (34.1)	NS
GIP (pmol/L)	132.3 (150.0)	95.3 (87.3)	122.2 (147.6)	NS
Triglyceride (mmol/L)	1.4 (0.6)	1.6 (0.8) *	1.1 (1.0)	<0.05
NEFA (mmol/L)	0.63 (0.29)	0.64 (0.23)	0.60 (0.26)	NS
HDL-C (mmol/L)	1.5 (0.4)	1.4 (0.3) *	1.7 (0.4)	<0.05
LDL-C (mmol/L)	3.3 (0.8)	3.3 (0.7)	3.1 (0.7)	NS
AST (U/L)	25.5 (8.8)	24.8 (6.3)	24.4 (8.0)	NS
ALT (U/L)	24.9 (10.2) *	22.5 (9.4) *	18.3 (6.3)	<0.05
γ-GTP (U/L)	50.9 (69.9)	40.0 (19.3)	36.5 (40.7)	NS

Values are means (standard deviation). Means were compared using the generalised estimating equations. Analyses were adjusted for total physical activity. Post-hoc analysis was adjusted for multiple comparisons using the Bonferroni method. Post-hoc analysis of the main effect of group: * *p* < 0.05 different from the non-obese group. ^†^
*p* < 0.05 different from the normal-weight obese group. HbA1c; hemoglobin A1c, HOMA-IR; homeostasis model assessment of insulin resistance, HOMA-β; homeostasis model assessment of β-cell function, GIP; glucose-dependent insulinotropic polypeptide, NEFA; non-esterified fatty acids, HDL-C; high-density lipoprotein cholesterol, LDL-C; low-density lipoprotein cholesterol, AST; aspartate aminotransferase, ALT; alanine aminotransferase, γ-GTP; γ-glutamyl transpeptidase, NS; Not significant.

**Table 3 ijerph-20-06408-t003:** Physical activity levels in the obese, normal-weight obese and non-obese groups.

	Obese Group (n = 27)	Normal-Weight Obese Group (n = 35)	Non-Obese Group (n = 81)	Main Effect of Group (*p* Value)
Steps (steps/day)	7431 (3015) *	8286 (2766)	9562 (3438)	<0.05
Total physical activity (min/day)	78 (31) *	86 (27)	98 (33)	<0.05
Light-intensity physical activity (min/day)	56 (23)	59 (19)	66 (23)	NS
Moderate-intensity physical activity (min/day)	21 (17)	26 (16)	31 (21)	NS
Vigorous-intensity physical activity (min/day)	1 (1) *	1 (1) *	2 (4)	<0.05

Values are means (standard deviation). Means were compared using the generalised estimating equations. Analyses were adjusted for age. Post-hoc analysis was adjusted for multiple comparisons using the Bonferroni method. Post-hoc analysis of the main effect of group: * *p* < 0.05 different from the non-obese group. NS; Not significant.

**Table 4 ijerph-20-06408-t004:** Three-day dietary data in the obese, normal-weight obese and non-obese groups.

	Obese Group (n = 27)	Normal-Weight Obese Group (n = 35)	Non-Obese Group (n = 81)	Main Effect of Group (*p* Value)
Energy intake (kJ/day)				
Total energy intake	9062 (1278)	9101 (1432)	9357 (1539)	NS
Breakfast	2226 (513)	2546 (712)	2356 (667)	NS
Lunch	2531 (614)	2549 (770)	2520 (700)	NS
Dinner	3192 (594)	3527 (993)	3479 (922)	NS
Snacks	910 (793)	702 (736)	966 (845)	NS
Food group (kJ/day)				
Cereals	3020 (621)	3008 (577)	3017 (662)	NS
Potatoes and starches	133 (128)	130 (101)	152 (124)	NS
Sugars and sweeteners	106 (60)	116 (79)	106 (81)	NS
Pulses	394 (321)	299 (234)	354 (275)	NS
Nuts and seeds	81 (29) *^,†^	102 (32)	104 (42)	<0.05
Vegetables	340 (121) *^,†^	427 (133)	436 (176)	<0.05
Fruits	391 (288)	347 (217)	371 (254)	NS
Mushrooms	6 (5)	9 (12)	9 (13)	NS
Algae	16 (23)	16 (40)	12 (15)	NS
Fish, mollusks and crustaceans	511 (347)	610 (400)	554 (303)	NS
Meat	898 (440)	984 (436)	985 (499)	NS
Eggs	340 (154)	289 (167)	273 (152)	NS
Milk and milk products	499 (407)	641 (481)	632 (448)	NS
Fats and oils	520 (246)	588 (277)	553 (247)	NS
Confectionaries	539 (539)	280 (346)	349 (416)	NS
Beverages	696 (535)	684 (712)	800 (864)	NS
Seasonings and spices	362 (161)	453 (239)	437 (220)	NS
Prepared foods	30 (78)	0 (0)	18 (73)	NS

Values are means (standard deviation). Means were compared using the generalised estimating equations. Analyses were adjusted for age. Post-hoc analysis was adjusted for multiple comparisons using the Bonferroni method. Post-hoc analysis of the main effect of group: * *p* < 0.05 different from the non-obese group. ^†^
*p* < 0.05 different from the normal-weight obese group. NS; Not significant.

## Data Availability

Not applicable.

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
