# Peer review of "Habitual Physical Activity and Dietary Profiles in Older Japanese Males with Normal-Weight Obesity"

_ijerph, 2023, doi:10.3390/ijerph20146408_

Round 1

Reviewer 1 Report

The study was to compare the characteristic of habitual physical activity and dietary pattern in Japanese aged men with obese group, normal weight obese group and no-obese group. The normal weight obese is defined as high in visceral fat subjects which is thought to be higher risk in metabolic diseases. It is worth to investigate the major caused of health risks which are physical activity and dietary pattern in the normal weight obese subjects. The current study showed normal weight obese group were higher in plasma LDL-c, TG and ALT compared with non-obese group. However, the physical activity and dietary pattern was failed to explain the differences of the lipids and liver function markers.

The authors were well interpreted the collected data and the manuscript was well prepared. Several minor correction is needed before publication.

1. p5, line 198-199. Please briefly describe the methods of the rest blood markers analysis.

2. p7, line 270, Table 3, the top column is misplaced. Please revise it.

3. p8, line 283, Table 4. For easy to read, please separate/put a gap between the energy intake and good group.

3. p8, line 283, Table 4. Some data of food group (g/day) looks unable to match the total energy intake. For example, 14 Fat and oil presented over 500g which would be over 4500 kcal. Please check it carefully.

4. Does the visceral obesity level influence the outcome?

Reviewer 2 Report

This study examined whether there are differences in habitual physical activity and diet between individuals with normal weight obesity and obese or non-obesity. Although there are several significant findings found among the three groups, this article is not reporting innovative findings and failed to explore the relationships between physical activity, dietary intake and functional tests, and blood biomarkers. In addition, there are plenty of typos throughout the manuscript and grammar mistakes. I suggest the author rewrite and proofread the manuscript carefully before submitting the manuscript to the journal.

1.     Page 1: typo, “BMI < 25 kg/m2 and visceral fat area (cm2) 100”

2.     Line 69: A participant flow diagram is shown in Figure 1.

3.     Some studies use the dominant hand for handgrip strength measurement, please justify the rationale using an average of both left and right hands for handgrip strength in this study.

4.     Line 128: Eliminate the dot. 10-second Open-Close Stepping Test (OCS-10)

5.     Lines 184-188, this long sentence has a grammar mistake and needs to be rewritten.

6.     The study only included men, please justify this and whether the conclusion can be translated into women or not.

7.     Visceral fat was measured using abdominal bioelectrical impedance analysis. Please specify this method from lines 96-97. Are you using the segmental BIA method for this? How reliable of this BIA method to predict visceral fat?

8.     What is the purpose of measuring OCS-10?

9.     For all tables, if it is labeled a symbol (e.g. */+, etc.), the main effect of the group should be <0.05. Not sure why the authors are repeating this in the last column.   

10.  In table 1, for the one-leg standing with vision (second) measurement, non-obese was significantly higher than the other two groups. Please explain this finding.

11.  Table 1, last item “imed Up and Go Test (second)” missed a “T”

12.  Line 24 of the abstract, “Lowed high-density lipoprotein cholesterol” should be corrected to “lowered”.

13.  Did the authors measure “Short Physical Performance Battery (SPPB)” which is a more robust measurement among the older population for lower extremity performances?

14.  Are there any correlations between physical activity and performance/functional tests? Do dietary intakes affect blood biomarkers? More data analysis should be done to explore the relationships among these 4 tables.

15.  The overall objective of the study was still unclear. Are authors simply reporting the data among these three groups, or exploring some correlations between physical activity, dietary intake and body composition and other biomarkers? Please justify the reasons why physical activity and dietary intake data were collected.

Reviewer 3 Report

This study addressed the clinical importance of normal-weight obesity in relation to its metabolic complications. The study is well prepared and conducted, and outcomes are well presented, and the Discussion is appropriate. 

Reviewer 4 Report

Thank you for the opportunity to review the topic „Habitual physical activity and dietary profiles in Japanese older men with normal weight obese“. This study examined whether there are differences in habitual physical activity and diet between individuals with normal weight obesity and obese or non-obesity.

Although this study is interesting and relevant, I have the following major concerns:

Introduction

I would suggest that the Introduction section could be started by presenting the problem of obesity in a global context. Can the Authors emphasize World Health Organization data?

I would suggest that terms such as “men“ and “women“ must be changed to “male“ and “female“ throughout the manuscript. Did the Authors perform gender analysis?

It seems necessary for Authors to justify why aged male were the target cohort of this study?

Materials and methods

It appears that the information in the methods section is not sufficient. The Authors must describe the following information in this section:

What type of research was carried out?

What the design was applied to the present study?

How was the representative sample size calculated?

What type of sampling procedure has been applied?

Line 79: The Authors did not performed an experiment (there was no any intervention). However, as shown in Figure 1, the subjects were randomized. Was this study as a case-control study in design?

The subsection “Statistical analyses“ must be overwritten. I propose that the Authors clearly specify dependent and independent variables. It is necessary to specify the all statistical tests have been used to test the hypothesis of this study. Describe any methods used to examine subgroups and interactions too.

Results

Table 1: p-value does not show the main effect. How was the effect size calculated? Maybe the Authors used other coefficients? The description of the study data in this table suggests that the ANOVA test has been used. However, all this information must be provided in the subsection of “Statistical analyses“. Additionally, I suggest that the Authors could calculate the ANOVA effect size (partial eta squared coefficients).

A similar problem exists for statistical analysis of empirical data in other tables too. Moreover, the Authors performed too many statistical tests in order to test the hypothesis of this study. The significance level (p-value) should therefore be adjusted by applying the Bonferroni correction.

Discussion

The limitations paragraph of the current study seems to have to be extended. Discuss limitations of the study, taking into account sources of potential bias or imprecision. Discuss both direction and magnitude of any potential bias, please.

The results of this study cannot be generalized. The conclusions of the study must also be written in a separate section.

On the other hand, the main findings of this study demonstrate the causality. Unfortunately, this study is just observational and the evidence derrived from main study findings can only appeal to the relationship between variables.

Finally, the Authors must make practical recommendations too.

Kind Regards

Round 2

Reviewer 2 Report

The author addressed most of the questions raised by the reviewer. Please make minor revisions before considering this article for publication.

1.     Query 7: Authors have addressed the methods from lines 112-117. Please also add limitations and/or reliability of BIA and in the discussion part.

2.     Query-8: please also insert some of your responses into the manuscript.

3.     Query 14&15: please also elaborate in the discussion part based on the authors responses.

4.     I suggest for the tables, only label P<0.05, and all other P>0.05, use NS to replace.

Reviewer 4 Report

The authors answered my questions,

However, I have some observations:

Minor concerns:

The terms “ES“ should be “d

Conclusions should be presented in a separate section

Major concerns:

The amount of overlapping text (percentage of similarity) must be reduced in this paper.

King Regards
